# Comparison of Performance between Single- and Multiparameter Luminescence Thermometry Methods Based on the Mn^5+^ Near-Infrared Emission

**DOI:** 10.3390/s23083839

**Published:** 2023-04-09

**Authors:** Tahani A. Alrebdi, Abdullah N. Alodhayb, Zoran Ristić, Miroslav D. Dramićanin

**Affiliations:** 1Department of Physics, College of Science, Princess Nourah Bint Abdulrahman University, P.O. Box 84428, Riyadh 11671, Saudi Arabia; 2Department of Physics and Astronomy, College of Science, King Saud University, P.O. Box 2455, Riyadh 11451, Saudi Arabia; 3Centre of Excellence for Photoconversion, Vinča Insitute of Nuclear Sciences—National Institute of the Republic of Serbia, University of Belgrade, P.O. Box 522, 11001 Belgrade, Serbia

**Keywords:** luminescence thermometry, luminescent materials, Mn^5+^, near-infrared luminescence

## Abstract

Herein, we investigate the performance of single- and multiparametric luminescence thermometry founded on the temperature-dependent spectral features of Ca_6_BaP_4_O_17_:Mn^5+^ near-infrared emission. The material was prepared by a conventional steady-state synthesis, and its photoluminescence emission was measured from 7500 to 10,000 cm^−1^ over the 293–373 K temperature range in 5 K increments. The spectra are composed of the emissions from ^1^E → ^3^A_2_ and ^3^T_2_ → ^3^A_2_ electronic transitions and Stokes and anti-Stokes vibronic sidebands at 320 cm^−1^ and 800 cm^−1^ from the maximum of ^1^E → ^3^A_2_ emission. Upon temperature increase, the ^3^T_2_ and Stokes bands gained in intensity while the maximum of ^1^E emission band is redshifted. We introduced the procedure for the linearization and feature scaling of input variables for linear multiparametric regression. Then, we experimentally determined accuracies and precisions of the luminescence thermometry based on luminescence intensity ratios between emissions from the ^1^E and ^3^T_2_ states, between Stokes and anti-Stokes emission sidebands, and at the ^1^E energy maximum. The multiparametric luminescence thermometry involving the same spectral features showed similar performance, comparable to the best single-parameter thermometry.

## 1. Introduction

The luminous properties of materials are profoundly influenced by temperature. Using a method known as luminescent thermometry, it is possible to determine the temperature of a substance if one monitors the properties of the substance being measured. In luminescence thermometry, the three features that are utilized most frequently are the emission intensity, lifetime, and form of the emission spectrum [1,2,3,4,5]. In the past decade, luminescence thermometry has made significant advancements, rapidly approaching the performance of competing technologies. This method offers various advantages over traditional ones. It can be used to monitor temperatures in hostile environments, in hard-to-reach places, on moving parts or surfaces, and where other methods may not be viable. In addition, it has a high spatial resolution and can measure temperatures at the microscale or nanoscale. As a result, luminescence thermometry has applications in numerous fields, such as materials science, nanotechnology, biology, and medicine.

However, luminescent thermometry has a number of drawbacks compared to other major thermometry techniques. The most serious are the decreased accuracy and precision, as well as the increased response time. With an accuracy of up to 0.1% of the measured temperature, thermocouples and resistance thermometers are regarded as providing the most precise temperature measurement [6,7]. In contrast, the accuracy of luminescence thermometry is often 0.5% or greater. The time resolution of thermocouples and resistance thermometers is normally measured in milliseconds or less [8,9], whereas those of steady-state luminescence thermometry are typically measured in the range of a few hundreds of milliseconds to seconds, and the ultrafast time-resolved measurements down to 10 ns are demonstrated using the excited state lifetime as a temperature indicator [10].

It is therefore understandable that a great deal of recent effort has been devoted to enhancing the accuracy and precision of luminescence thermometry. To achieve this objective, various strategies have been implemented. These include the selection of luminescent materials with both high brightness and high temperature sensitivity. Optical and electronic design enhancements to the luminescence thermometry system can also affect measurement precision and accuracy. Environmental control measures, such as thermal insulation or light shielding, can reduce the effects of these variables and enhance the precision and accuracy of measurements. It has been demonstrated that the identification of luminescence features with a high sensitivity to temperature fluctuations over a specific temperature range can enhance thermometric performance. For instance, the incorporation of high-energy thermalized levels for the realization of trivalent lanthanide Boltzmann thermometers may result in a substantial increase in the relative sensitivity [11,12,13,14,15]. In dual-excited single-band ratiometric thermometry, the greater energy difference between the lower energy states of trivalent lanthanides enhances the relative sensitivity [16]. Recently, a combination of these two approaches has increased the relative sensitivity even further [17]. By utilizing time-resolved measurements, the LIR method can be advanced further. Qiu et al. [18] demonstrated that the luminescence intensity ratio methodology can be improved if performed with time-resolved measurements, in which the ratio of emission intensities for the same emission band are measured at different time-delays after excitation.

The great majority of luminescence thermometers that have been investigated so far rely on a single parameter to deliver an accurate temperature reading. The utilization of multiple thermometric parameters within the context of multimodal [19,20,21,22,23] or the multiparametric approach [24,25,26,27,28], on the other hand, has the potential to enhance the performance of thermometers. So far, the advantage of multiparametric thermometry has been realized through the application of multiple linear regression or similar statistical techniques and artificial neural networks [10,29,30]. The basic assumption of multiparametric thermometry is that the involvement of multiple temperature readings will accumulate sensitivity and cancel individual deviations in measurements by the process of averaging. If there are no artifacts in the measurements, then all the thermal readings should converge. Nevertheless, the inclusion of additional spectral feature measurements increases the uncertainty in temperature determination, which may negate the benefits obtained.

The purpose of this study was to compare the performance of single- and multiparametric temperature measurements derived from identical emission spectrum measurements. For this purpose, we chose the Mn^5+^ emission, which provides an abundance of temperature-dependent spectral features, such as intensity ratios between emissions from coupled excited states or between Stokes and anti-Stokes sidebands, and the shift of an emission band. Additionally, Mn^5+^-activated luminescent phosphors and nanophosphors emit in the near-infrared spectral region, at wavelengths greater than 1000 nm, where environmental background luminescence is minimal. This facilitates the use of these nanophosphors in biomedical applications, for instance [31].

## 2. Materials and Methods

For the synthesis of Ca_6_BaP_4_O_17_:Mn^5+^ (0.75 at% Mn) material, the typical solid-state reaction was utilized, as described in [32]. CaCO_3_ (Alfa Aesar, 98%), BaCO_3_ (Alfa Aesar, 99.8%), (NH_4_)H_2_PO_4_ (Alfa Aesar, 98%), and MnO (Aldrich, 99.99%) were weighed and thoroughly mixed for one hour in an agate mortar with the required amount of ethanol. Raw material mixtures were poured in alumina crucibles and heated in an air atmosphere at 600 °C for six hours before being ground in an agate mortar and calcinated at 1200 °C for an additional ten hours. The procedure yielded a powder of a turquoise color composed of micron-size crystallites (see the scanning electronic microscopy image in Appendix A) which perfectly corresponds to the Ca_6_BaP_4_O_17_:Mn^5+^ crystal structure (see the powder X-ray diffraction pattern in Appendix A). Photoluminescence emission spectra were measured in the 20 °C to 100 °C temperature range using a custom-made Peltier-based heating stage with temperature monitoring using a four-wire PT100 sensor (PICO Technology PT-104; 1/10 DIN accuracy of 0.03 °C; 0.002 °C precision). Excitation is provided by an Ocean Insight LSM-635A fiber coupled LED (2.68 mW maximum power), which is operated by a single-channel Ocean Insight LDC-1 driver and controller. An Ocean Insight NIRQuest+ spectrometer was connected by the bifurcation optical Y cable for PL emission spectrum measurements. With 350 ms of integration time per scan and five averaged scans for each spectrum, the total acquisition time per spectrum was 1.75 s. A field-emission scanning electron microscope (FE-SEM) Tescan Mira 3 XMU operated at 10 keV was used to observe the morphology of the samples. Prior to SEM analysis, samples were sputter-coated with a thin layer of gold to ensure conductivity. The crystal structure of powders was examined by powder X-ray diffraction using the Rigaku SmartLab instrument (Cu-Kα_1,2_ radiation; λ = 0.1540 nm) at room temperature. Data were recorded over the 10°−90° 2θ range, with a 0.01° step size and 1 min/° counting time. Using the license granted to the University of Belgrade, built-in MATLAB 2021b functions were used to perform single- and multiparametric linear regressions on collected data.

## 3. Results

### 3.1. Photoluminescence and Single-Parameter Thermometry

Under the 635 nm excitation into the broad ^3^A_2_ → ^3^T_1_ electric dipole-allowed transition, the Mn^5+^ emits over the 7500–10,000 cm^−1^ NIR spectral range, see Figure 1a. Its emission spectrum is composed of the narrow emission band that originates from the spin-forbidden ^1^E → ^3^A_2_ intraconfigurational transition centered around 8772 cm^−1^ (1140 nm), the low-intensity broad emission from the spin-allowed ^3^T_2_ → ^3^A_2_ transition, and Stokes and anti-Stokes vibronic sidebands distanced from ^1^E → ^3^A_2_ emission peak at approximately 320 cm^−1^ (from O—Mn—O bending) and 800 cm^−1^ (from Mn—O stretching) [32]. The shape of the emission spectrum is significantly influenced by temperature fluctuations in a number of ways. The intensity of ^1^E emission decreases as the temperature rises, while the intensity of ^3^T_2_ emission increases. Because the difference in their energies is approximately 1260 cm^−1^ [32], these two excited states are in thermal equilibrium at room temperature and share their respective populations. The intensity of Stokes and anti-Stokes emission bands exhibits a similar temperature-dependent trend. While the former decreases in intensity as the temperature rises, the latter increases in intensity. At elevated temperatures, the spectral position of the ^1^E emission band red-shifts and broadens markedly.

In luminescence thermometry, each of these temperature-dependent spectral features can be used independently as temperature indicators [2]. For luminescence thermometry, the luminescence intensity ratio (LIR) between two emission bands (so-called ratiometric temperature readout) has been and remains the most popular method [33]. It is of the utmost importance because it is self-referencing and therefore circumvents problems caused by changes in measurement conditions by relying on ratios of absolute intensities. Considering the composition of the spectra depicted in Figure 1a, one can exploit two luminescence intensity ratios that both follow the Boltzmann distribution, which gives the population probability of a particular state as a function of that state’s energy and temperature. The first intensity ratio, denoted in the text as *LIR*_1_, can be formulated from emission intensities of the ^1^E and ^3^T_2_ states. Two excited energy states are considered thermally coupled when they are close in energy and when the nonradiative transition rates between them surpass the radiative transition rates within the considered temperature range [34].

Then, we may consider these states are in the Boltzmann equilibrium and the intensity ratio of their emissions has the following mathematical representation [35]:(1)$${LIR}_{1}(T)=B\cdot exp(-∆E/kT),$$
where ∆*E* is the energy difference between the thermally coupled emitting states (here, the ^1^E and ^3^T_2_ states), *k* = 0.695 cm^−1^ is Boltzmann’s constant, *B* is a temperature-invariant constant, and *T* represents absolute temperature. To avoid deconvolution of overlapping emission bands [36], *LIR*_1_ was calculated from integrated intensity of emission near the ^1^E band maximum and the integrated emission of the part of the emission spectrum corresponding to the ^3^T_2_ emission that does not overlap with ^1^E and anti-Stokes emissions (*E* > 9700 cm^−1^). Each spectrum at every temperature was submitted to the baseline removal procedure that consisted of utilizing part of the spectrum with no Mn^5+^ emission as a baseline. The temperature dependence of *LIR*_1_ (symbols on Figure 1b) perfectly fits Equation (1) (full line; R^2^ = 0.99998), with the value of ∆*E* = 1216 cm^−1^ that is in good agreement with the energy difference between ^1^E and ^3^T_2_ of 1260 cm^−1^ [32].

The second intensity ratio, denoted in the text as *LIR*_2_, can be constructed from the ratio of intensities of Stokes and anti-Stokes vibronic sidebands that arise from the coupling of the ^1^E excited state and the non-totally symmetric O—Mn—O bending deformation [29]. To analyze the temperature behavior of Stokes and anti-Stokes emissions, the baseline was removed for each spectrum by fitting third-order polynomials on the emission data in the vicinity of the corresponding peaks (see Appendix A). The values of such obtained polynomials inside the corresponding peaks were then used as a baseline. Although the experimental dependance of *LIR*_2_, Figure 1c, can be fitted with the Boltzmann distribution formula, Equation (1), where we obtain ∆*E* = 330.4 cm^−1^ which agrees well with the energy separation of the first vibronic side band of 320 cm^−1^, we found that much better regression with the simple second-order polynomial expression (full line in Figure 1c; R^2^ = 0.9993; parameters obtained after the fitting are given in Table 1):(2)$${LIR}_{2}\left(T\right)=a\cdot T^{2}+b\cdot T+c.$$

The redshift of the ^1^E emission band upon temperature increase is shown in Figure 1d, while the temperature dependence of the ^1^E emission band maximum (the peak energy) is given in Figure 1e with symbols. The position of this peak can be well approximated by a linear equation over the complete measurement range (full line; R^2^ = 0.9994; parameters obtained after the fitting are given in Table 1):(3)$$E_{1E}\left(T\right)=d\cdot T+h.$$

At room temperature, the relative sensitivities of *LIR*_1_ and *LIR*_2_ are approximately 2%K^−1^ and 0.3%K^−1^, respectively, while the peak energy changes at the 0.21 cm^−1^K^−1^ rate.

### 3.2. Multiparametric Luminescence Thermometry

Multiparametric linear regression (MLR) is a statistical method that utilizes two or more explanatory variables to determine the outcome of a response variable. Each value of the independent variable is associated with a value of the dependent variable. Using multiple linear regression, one can represent the linear relationship between the thermometric parameters Δ (independent variables) and temperature (dependent variable) using the following equation:(4)$$T=\beta_{0}+\beta_{1}\Delta_{1}+\beta_{1}\Delta_{1}+\beta_{1}\Delta_{1}+\cdots +\beta_{n}\Delta_{n}+\varepsilon ,$$
where *β*_0_ is the constant term (*T*-intercept), *β*_1_, …, *β*_n_ are the slope coefficients for each of *n* explanatory variables, and ε is the model’s error term (also known as the residuals).

The prerequisite for using MLR is that there is a linear relationship between the dependent variables and the independent variables and that the independent variables are not too highly correlated with each other. Although it is not strictly required, the values of independent variables should be in the same range (the same magnitude) since fitting procedures assume that larger values have more significance. This can be achieved by feature scaling. An additional benefit of feature scaling is that it makes interpretation of results much easier since it makes a lot more sense to say one feature is more important than another if they are of the same magnitude when considered.

Temperature indicators in luminescence thermometry, such as the emission intensity, luminescence intensity ratio, excited state lifetimes, band shifts, bandwidths, etc., are not linear functions of temperature. Therefore, they cannot be straightforwardly used in MLR, although the successful application of MLR can be demonstrated over the limited temperature range in which the linearity can be assumed [24,25].

Here, we propose a procedure for obtaining explanatory values from the above-mentioned temperature indicators, which provide a linear relationship with a response variable. In the first step, the equation of state should be formulated for each of the temperature indicators [30]. For the luminescence intensity ratios and the shift of emission band maximum used in this work, this means the respective transformation of Equations (1)–(3) to the following forms:(5)$$T_{{LIR}_{1}}=\frac{\Delta E}{k\cdot \left(\log\left(B\right)-\log\left({LIR}_{1}\right)\right)}=\Delta_{1},$$
(6)$$T_{{LIR}_{2}}=\frac{-b+\sqrt{b^{2}-4a\cdot \left(c-{LIR}_{2}\right)}}{2a}=\Delta_{2},$$
(7)$$T_{E_{1E}}=\frac{E_{1E}-h}{d}=\Delta_{3}.$$

Then, the linearized thermometric parameters Δ are calculated using parameters obtained from single-parametric regressions (Table 1). This approach also provides a form of feature scaling, as the resulting thermometric parameters have temperature units and so have values of the same magnitude. The multiparametric linear regression with linearized and scaled explanatory variables Δ was implemented effectively.

### 3.3. Performance Comparison of Multiparametric Luminescence Thermometry with Single-Parameter Thermometry

To compare the performance of three single-parameter thermometry methods and the multiparametric method, we performed an additional 50 measurements at each temperature and calculated the accuracy (the extent of agreement between a measured temperature value and the nominal temperature) and precision (the standard deviation of repeated temperature measurements) for each method and temperature. The results are shown on Figure 2a,b, respectively. Figure 2d–g show the distribution of calculated temperatures at the nominal temperature of 328.15 K, for *LIR*_1_, *LIR*_2_, $E_{1E}$ based single-parametric methods, and the multiparametric method that utilizes these three thermometric indicators, respectively.

The accuracy of multiparametric thermometry is much better than the accuracies of *LIR*_2_- and *E*^1^_E_-based single-parametric thermometry, but it is almost the same over the complete temperature range as the accuracy of *LIR*_1_ thermometry. The accuracy values averaged for all temperatures (Table 2) show almost the same values obtained by multiparametric thermometry and *LIR*_1_. The precision of multiparametric thermometry is slightly worse than the one obtained with *E*_1E_, similar to that obtained with *LIR*_1_, and better than precision of *LIR*_2_ (Figure 2b; Table 2). In this case, it appears that linear multiparametric regression resembles the quality of the best linear regression of the explanatory variable involved in the multiparametric fit.

To test this hypothesis, we ran three multiparametric regressions using two explanatory factors with three variable combinations, i.e., three two-parameter regressions. Figure 3 compares the accuracy and precision obtained from two-parameter and single-parameter regressions. As in the case with three-parameter linear regression, the accuracies obtained with two-parameter regressions resembled that of the best linear regression involved, whereas the precisions were the same or slightly worse.

## 4. Discussion

Despite the fact that the spectrum of Mn^5+^ near-infrared emission comprises numerous features with significant temperature dependences, only a few of them can be employed for accurate and precise thermometry. They are the luminescence intensity ratio between ^1^E and ^3^T_2_ emissions, which offers accuracy of 0.09 K, and the energy of the ^1^E band, which provides 0.16 K precision. Using multiparametric thermometry for further large gain in accuracy and precision seems unlikely. This is consistent with the observation that the relative sensitivity of a linear multiparametric thermometry is equal to the sum of the relative sensitivities of each involved single-parameter thermometry multiplied by slope coefficients [24,25]. Since the sum of all slope coefficients is one, the relative sensitivity of a linear multiparametric thermometry cannot exceed the greatest relative sensitivity of a single-parameter thermometry. Given that the uncertainty in the determination of more thermometric parameters is always greater than the uncertainty in the determination of a single parameter, the uncertainty in multiparametric thermometry should be larger than in the best performing single-parameter thermometry, as it is defined as a ratio of total parameter uncertainty to total relative sensitivity.

One should note that the accuracy and precision of luminescence thermometry are not its intrinsic properties since they are significantly affected by the spectrometer’s quality and resolution [37]. Considering that spectral measurements in this study were taken with a simple portable instrument, it is reasonable to assume that a more advanced spectrometer would result in improved thermometry performance.

Considering that the obtained inputs for multiparametric regression are temperatures, the demonstrated linearization of explanatory values additionally provides for feature scaling. When the input values of a linear multiparametric regression are of equal magnitude, the fit gives them equal weight. In principle, the linearization does not need to be based on the established physical model, as in Equation (1), but rather on the optimal regression, as in Equations (2) and (3), as the quality of these regressions will affect the quality of multiparametric regression.

The Mn^5+^ emission spectrum provides additional temperature-dependent features not used in this study for two reasons. The LIR between ^1^E emission and Stokes (or anti-Stokes) emission and the LIR between Stokes and anti-Stokes emission from Mn—O stretching sidebands are not included in multiparametric regression since they are correlated with two already included LIRs. They either involve the same intensity of emission (^1^E emission band), the same phenomenon (Boltzmann distribution for Stokes and anti-Stokes intensities), or the same two-phonon Raman process that is responsible for both the energy band shift and broadening. In contrast, the excited state lifetime is a suitable input for multiparametric thermometry. It is not strongly correlated with either emission ratios or energy band shift, and it is known as one of the principle single-parameter luminescence thermometry methods. However, the main aim of this analysis was to compare the performance of single- and multiparametric temperature readings from the same steady-state spectrum. Since a separate time-resolved measurement is required for the lifetime, it was not suited for this purpose.

## 5. Conclusions

In conclusion, the precision and accuracy of single- and multiparametric luminescence thermometry based on the steady-state near-infrared emission of Mn^5+^ were compared. We initially measured 50 emission spectra of Ca_6_BaP_4_O_17_:Mn^5+^ phosphor over the temperature range of 293–373 K in 5 K increments. Observing that a number of spectral features vary with temperature fluctuations, we chose luminescence intensity ratios between emissions from the ^1^E and ^3^T_2_ states, luminescence intensity ratios between Stokes and anti-Stokes emission sidebands, and ^1^E energy as temperature indicators for three single-parameter thermometry readouts. Then, we independently calculated the accuracy and precision of each of these methods and incorporated them into the multiparameter readout. We showed that linearization and feature scaling of input parameters for linear multiparametric regression may be achieved by deriving the expression for temperature from the existing physical model or from the best regression equation. The choice between two should be determined by the quality of single-parameter regressions, as they impact the performance of multiparameter regression. Our analysis demonstrates that the accuracy and precision of the multiparametric output are nearly identical to those of the best performing single-parameter or multiparametric regression method. Whether it is a general rule is yet to be resolved in future work with other combinations of temperature readouts and other luminescent centers. The precision of multiparametric temperature readings may slightly worsen compared to the most precise single-parametric readings, likely due to the uncertainty accumulation caused by the larger number of spectral parameters used in multiparametric temperature readings.

## Figures and Tables

**Figure 1 sensors-23-03839-f001:**
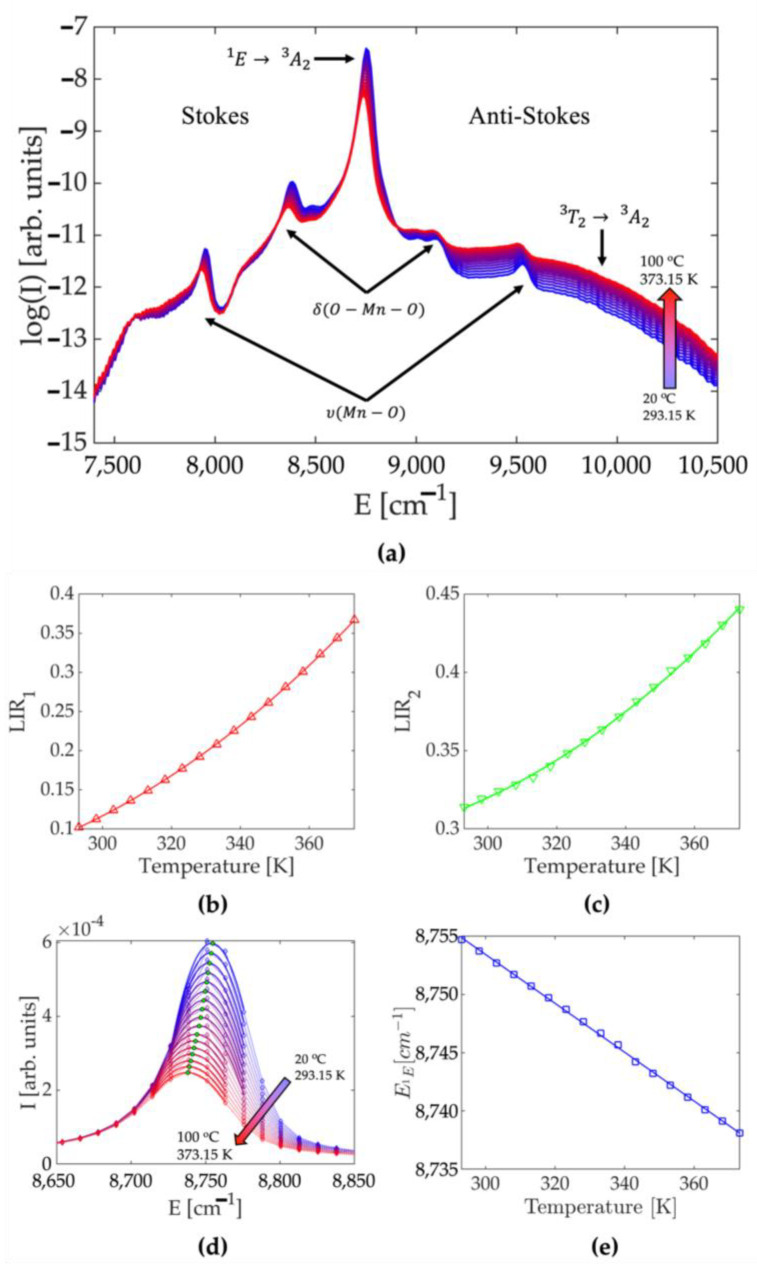
(**a**) Photoluminescence emission spectra of Ca_6_BaP_4_O_17_:Mn^5+^ phosphor measured at different temperatures (excitation at 635 nm); (**b**) temperature dependence of the luminescence intensity ratio of ^1^E and ^3^T_2_ emissions (*LIR*_1_; experimental data are represented with symbols; fit to Equation (2) is given by full line); (**c**) temperature dependence of the luminescence intensity ratio of Stokes and anti-Stokes sidebands (*LIR*_2_; experimental data are represented with symbols; fit to Equation (3) is given by full line); (**d**) the redshift of the ^1^E emission band upon temperature increase; (**e**) ^1^E emission band maximum variation with temperature (*E*_1E_; experimental data are represented with symbols; fit to Equation (4) is given by full line).

**Figure 2 sensors-23-03839-f002:**
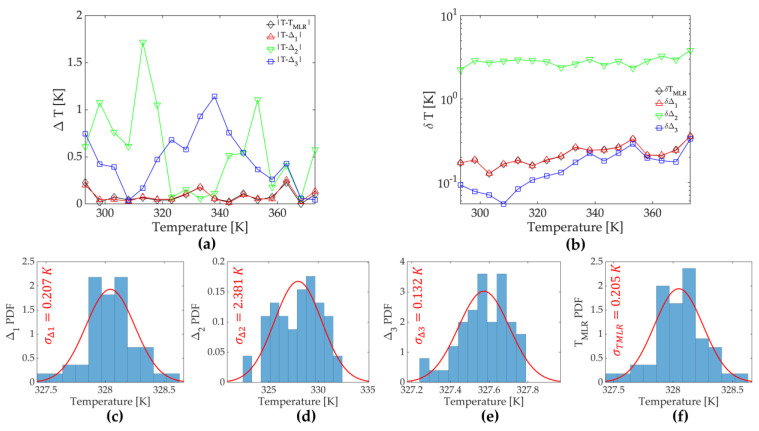
(**a**) The accuracy (Δ*T*) and (**b**) the precision (δ*T*) of single-parametric (*LIR*_1_—red ∆ symbol and line, *LIR*_2_—green ∇ symbol and line, and *E*_1E_—blue □ symbol and line) and multiparametric temperature (black ♢ symbol and line) readings from the near-infrared emission of Ca_6_BaP_4_O_17_:Mn^5+^ at different temperatures; (**c**–**f**) distributions of temperatures measured by single-parametric (*LIR*_1_, *LIR*_2_, and *E*_1E_) and multiparametric thermometry, respectively (nominal temperature 328.15 K); temperature distributions were fitted to the normal distribution (red line), and obtained standard deviations are written on graphs (note that a full width at half maximum (FWHM) of distribution can be obtained from a standard deviation as $FWHM=2\cdot \sqrt{2\cdot log(2\sigma )}\approx 2.355\sigma$ ).

**Figure 3 sensors-23-03839-f003:**
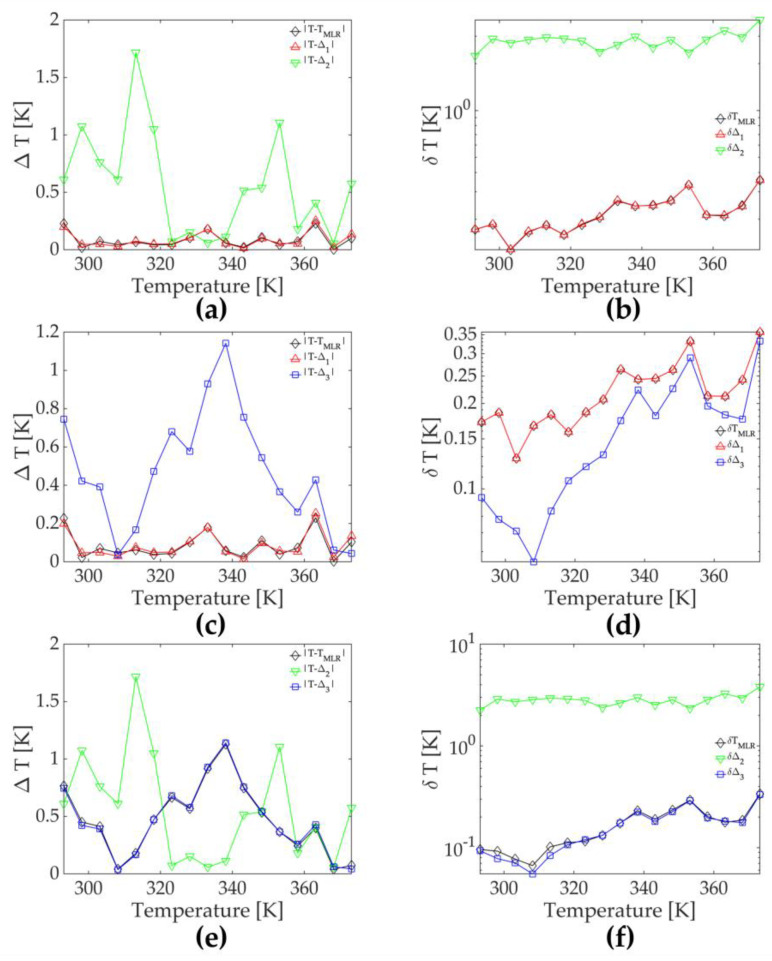
(**a**) The accuracy (Δ*T*) and (**b**) the precision (δ*T*) of single-parametric (*LIR*_1_—red ∆ symbol and line, and *LIR*_2_—green ∇ symbol and line) and two-parameter (black ♢ symbol and line) temperature readings at different temperatures; (**c**) the accuracy (Δ*T*) and (**d**) the precision (δ*T*) of single-parametric (*LIR*_1_—red ∆ symbol and line, and *E*_1E_—blue □ symbol and line) and two-parameter (black ♢ symbol and line) temperature readings at different temperatures; (**e**) the accuracy (Δ*T*) and (**f**) the precision (δ*T*) of single-parametric (*LIR*_2_—green ∇ symbol and line, and *E*_1E_—blue □ symbol and line) and two-parameter temperature (black ♢ symbol and line) readings at different temperatures.

**Table 1 sensors-23-03839-t001:** The parameters obtained by fitting experimental results to Equations (2)–(4).

Parameters from Fitting	*B*	*∆E* [cm^−1^]	*a* [K^−2^]	*b* [K^−1^]	*c*	*d* [K^−1^cm^−1^]	*h* [cm^−1^]
	39.93	1216	8.667 × 10^−6^	−4.174 × 10^−3^	7.921 × 10^−1^	−2.098 × 10^−1^	8.816 × 10^3^

**Table 2 sensors-23-03839-t002:** The accuracy (Δ*T*) and precision (δ*T*) of single-parametric (*LIR*_1_, *LIR*_2_, and *E*_1E_) and multiparametric temperature readings from the near-infrared emission of Ca_6_BaP_4_O_17_:Mn^5+^ (0.75 at% Mn) averaged over the complete temperature range.

Method	*LIR* _1_	*LIR* _2_	*E* _1*E*_	MLR
Δ*T* [K]	0.0868	0.5649	0.4720	0.0841
δ*T* [K]	0.2213	2.8204	0.1601	0.2212

## Data Availability

The data presented in this study are available in Appendix A (‘ALLDATA.txt’).

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
