# Peer review of "Comparison of Performance between Single- and Multiparameter Luminescence Thermometry Methods Based on the Mn5+ Near-Infrared Emission"

_sensors, 2023, doi:10.3390/s23083839_

Round 1
Reviewer 1 Report
1. The SEM image of synthesized phosphor should be added.
2. XRD should be added to confirm your crystalline phase.
3. I cannot understand why the Mn ions exist in the type of Mn5+?
4. Your R2 = 0.99998, please explain that.
Author Response
- The SEM image of synthesized phosphor should be added.
As suggested, the SEM image of synthesized phosphor is added as Figure S1.
- XRD should be added to confirm your crystalline phase.
As suggested, the XRD is added as Figure S2.
- I cannot understand why the Mn ions exist in the type of Mn5+?
Manganese exhibits oxidation states from +2 to +7. In Mn5+ oxidation state it has 3d2 electron configuration. It is easily stabilized in this oxidation state in this material since it replaces P5+ which has the same valance state.
https://www.sserc.org.uk/subject-areas/chemistry/chemistry-resources/oxidation-states-of-manganese/
- Your R2 = 0.99998, please explain that.
This indicates almost perfect fit of experimental data to theoretical model.

Reviewer 2 Report
Your effort to investigate use of multiple parameters for thermometry resulted in the less than exciting finding that it was no better than using the single parameter with the best accuracy. So I find the novelty and significance to be low. The way the data is manipulated is also unclear. This could be repaired to the point that the reader would be assured that you did things like handling baseline subtractions and noise appropriately. There are a few additional questions and comments about specific parts of the manuscript in the attached pdf file. But overall I do not think this manuscript, even after making suggested changes, would be appropriate for publication in this journal.

Author Response
Thank you for your valuable comments. We amended the manuscript in accordance with the report's and pdf's suggestions.
- We rewrite text so that readers can easily understand how data is manipulated and how baseline subtraction was done.
- We corrected the figures based on your suggestions and included standard deviation and full-width at half-maximum (FWHM) values for the normal distributions shown in Figure 2.
- We included a new panel in Figure 1 to clearly show the red shift of the 1E emission band upon temperature increase.
- We included the missing references.
- We included complete information on how temperature was measured in the heating stage, with accuracy and precision data.
- We removed the term ‘strong’ when referring to the emission from a spin-forbidden transition.
- We clarified that precision isn't just the noise in our measurement since we used integrated emissions for intensities. Then, the precision is obtained as a standard deviation of 50 measurements.
- We provided all measured spectra in an open access form.
- We corrected all unambiguous terms and typing errors.
I disagree with your evaluation of the significance and originality of our results and analysis. The result indicating that linear multiparametric regression luminescence thermometry is not superior in terms of accuracy and precision to the single parameter thermometry with the highest accuracy is significant for future research in the field of luminescence thermometry, given that there is evidently ongoing research on this topic and that recent papers on this topic have not addressed this issue. Certainly, we make conclusions based on this single event, and we acknowledged as much in the amended publication with the comment that more research is needed to determine whether or not this is a general case. For your information, we analyzed a number of additional cases involving different types of luminescence centers and thermometric parameters, all of which support the presented conclusion.
In addition to the examination of the accuracy and precision of the linear multiparametric and conventional approaches, we also presented the methodology for employing the linear multiparametric approach in luminescent thermometry. Previous publications did not address linearization and feature scaling of input data, despite the fact that they are necessary for the use of linear multiparametric regression, which is well-known in other disciplines. For instance, without feature scaling, the regression will assign more weights to inputs with greater magnitudes. In addition, the suggested scaling of features yields dimensionless beta coefficients that may be compared to determine the relative significance of individual inputs. As temperature dependences of luminescence properties are typically nonlinear functions, linearization of input data is of the utmost importance. Without it, the multiparametric technique can only be applied to short temperature ranges where linearity may be assumed.

Round 2
Reviewer 1 Report
Accept in present form
Author Response
Thank you for your positive assessment.
Reviewer 2 Report
The authors have answered my questions and comments appropriately. While I still consider the significance of the manuscript low, it may be of interest to others working in the field. So I will support publication.
Author Response
Thank you for your positive assessment.